# Dialogizer: Context-aware Conversational-QA Dataset Generation from Textual Sources

**Yerin Hwang**[1*]  **Yongil Kim**[2*]  **Hyunkyung Bae**[3]
**Hwanhee Lee**[4†]  **Jeesoo Bang**[3]  **Kyomin Jung**[1,2,5†]

[1]IPAI, Seoul National University  [2]Dept. of ECE, Seoul National University
[3]LG AI Research  [4]Chung-Ang University  [5]SNU-LG AI Research Center
{dpfls589, miles94, kjung}@snu.ac.kr
{hkbae, jeesoo.bang}@lgresearch.ai, hwanheelee@cau.ac.kr

## Abstract

To address the data scarcity issue in Conversational question answering (ConvQA), a dialog inpainting method, which utilizes documents to generate ConvQA datasets, has been proposed. However, the original dialog inpainting model is trained solely on the dialog reconstruction task, resulting in the generation of questions with low contextual relevance due to insufficient learning of question-answer alignment. To overcome this limitation, we propose a novel framework called **Dialogizer**, which has the capability to automatically generate ConvQA datasets with high contextual relevance from textual sources. The framework incorporates two training tasks: question-answer matching (QAM) and topic-aware dialog generation (TDG). Moreover, re-ranking is conducted during the inference phase based on the contextual relevance of the generated questions. Using our framework, we produce four ConvQA datasets by utilizing documents from multiple domains as the primary source. Through automatic evaluation using diverse metrics, as well as human evaluation, we validate that our proposed framework exhibits the ability to generate datasets of higher quality compared to the baseline dialog inpainting model.

## 1 Introduction

Dialog systems (Huang et al., 2020b; Ni et al., 2023) are designed to engage in natural language conversations with users, provide relevant information, answer queries, and simulate human interactions. These systems have gained significant attention in both academics and industry owing to their various potential applications, such as online customer service, virtual assistants, and interactive chatbots (Jia, 2004; Ghose and Barua, 2013; Nuruzzaman and Hussain, 2020). However, the scarcity of datasets poses a major challenge in

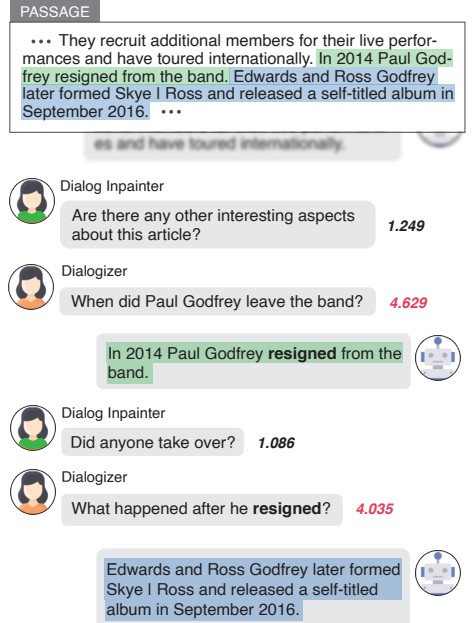

Figure 1: Example of questions generated by the proposed framework, Dialogizer. Dialogizer excels at generating contextually relevant questions compared to the original dialog inpainter. R$Q$UGE (Mohammadshahi et al., 2022) scores are provided on the right side.

the development of dialog systems. In particular, for information-seeking conversational question-answering (ConvQA) tasks (Stede and Schlangen, 2004; Zaib et al., 2022), creating a high-quality domain-specific dataset is costly because it requires the direct involvement of domain experts in data annotation (Demszky et al., 2021).

Recent work (Dai et al., 2022) proposes a dialog inpainting method to address this challenge by automatically generating ConvQA datasets using pre-existing text datasets. The text dataset is segmented into sentence-level units, which are directly utilized as answers, while the trained dialog inpainter generates questions corresponding to these answers to complete the conversation. Dialog inpainting has the potential to address the data scarcity issue owing to the abundance of online documents authored by domain experts and the capability to convert

---

*Equal Contribution.
†Corresponding authors.

these documents into dialogs with well-defined answers. Considering the guaranteed quality of the answers, it is crucial to generate questions that are well-aligned with each corresponding answer (Sun et al., 2018). However, we have observed a low contextual relevance in the questions generated by the dialog inpainter trained solely on a dialog reconstruction task, as it lacks sufficient training of question-answer alignment. For instance, as illustrated in Figure 1, the dialog inpainter tends to generate questions that exhibit a deficiency in answer specificity (e.g., the green case) or are contextually inappropriate (e.g., the blue case). Through experimental analysis, we quantitatively demonstrate that the questions generated by the dialog inpainter have low R$Q$UGE (Mohammadshahi et al., 2022) scores, indicating a lack of contextual relevance.

This study introduces Dialogizer as a novel framework for generating contextually relevant ConvQA datasets from textual datasets. The framework incorporates two training methodologies, in addition to dialog reconstruction, to address the limitation of generating contextually-irrelevant questions. These methodologies include a question-answer matching (QAM) task and a topic-aware dialog generation (TDG) task. In the QAM task, the model is provided with numerous QA pairs and learns to differentiate between matching and non-matching pairs. This enables the model to discern contextual relevance among QA pairs. In the TDG task, we provide the model with keywords extracted from the target answer using a keyword extractor. Then, the model learns to generate answer-specific questions using these keywords along with the given answer sentence and the dialog history. By incorporating these training tasks, the model becomes capable of generating more specific and answer-relevant questions. Furthermore, during the inference process of generating dialog from the passage, re-ranking is conducted using contextual relevance as a metric, ensuring the generation of high-quality questions.

Using Dialogizer, we compile four ConvQA datasets by leveraging source documents from various domains, such as news and medicine. Through automatic evaluation using multiple reference-free dialog metrics, human evaluation, GPT-4 evaluation (Chiang and Lee, 2023; Liu et al., 2023), and an application to text retrieval tasks, we experimentally demonstrate that our Dialogizer-generated datasets exhibit higher quality and context rele-

vance than those generated by the vanilla dialog inpainter. Furthermore, we present an ablation study that shows the effectiveness of each methodology of the proposed framework. Our results represent evidence supporting the potential impact of the Dialogizer in advancing ConvQA research.

## 2 Related Works

### 2.1 Conversational Question-Answering

Conversational question-answering (ConvQA) aims to enable machines to effectively answer multiple questions from users based on a given passage. ConvQA datasets must contain accurate information regarding specific domains, maintain consistency across topics, and facilitate the progression of dialog turns hierarchically (Zhu et al., 2021). However, creating ConvQA datasets is labor-intensive, as evidenced by the manual annotations throughout existing ConvQA datasets such as CoQA (Reddy et al., 2019), CSQA (Saha et al., 2018), and ConvQuestions (Christmann et al., 2019). In this work, we present a framework designed to automatically generate high-quality ConvQA datasets and provide empirical evidence that generated datasets may serve as valuable resources for ConvQA tasks.

### 2.2 Dialog Inpainting

To overcome the data scarcity problem in ConvQA, an automatic ConvQA dataset generation framework called dialog inpainting (Dai et al., 2022) has recently been developed. Similar to filling in one side of a phone call conversation by overhearing the other side, this methodology considers sentences from text documents as one person's utterances to generate the remaining utterances, thereby completing the conversation. The efficiency of dialog inpainting as a ConvQA dataset generation framework is demonstrated by its ability to automatically convert text data into a dialog format without loss of information, as evidenced by the abundance of high-quality documents annotated by experts in domains. However, we have experimentally observed that the questions generated via dialog inpainting lack contextual relevance. In this study, we propose Dialogizer as a framework for generating contextually relevant ConvQA datasets.

### 2.3 Reference-free Dialog Metrics

Owing to the one-to-many nature of dialog and question generation tasks (Zhao et al., 2017), reference-based natural language generation met-

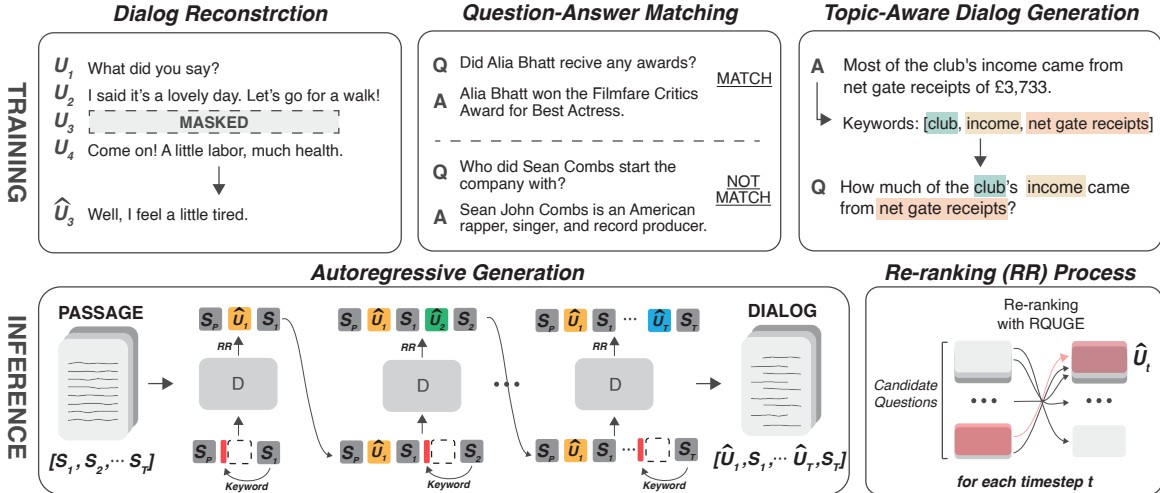

Figure 2: An overview of the proposed Dialogizer framework. In the training phase, in addition to the dialog reconstruction task, two novel tasks are incorporated: Question-Answer Matching and Topic-aware Dialog Generation. During the inference phase, autoregressive question generation is performed using textual data to complete the conversation, and for each question generation, re-ranking is conducted based on the contextual relevance.

rics (Papineni et al., 2002; Lin, 2004) have shown a poor association with human judgment (Liu et al., 2016; Lowe et al., 2017; Gupta et al., 2019). Additionally, these metrics have a limitation as they can only be applied in situations where a reference is available. Therefore, recent studies have focused on developing reference-free metrics (Huang et al., 2020a; Gao et al., 2020; Zhang et al., 2021) that exhibit high correlations with human judgment. Hence, we employ various reference-free metrics to evaluate the quality of generated ConvQA datasets. In addition, the reference-free metrics can be used as re-ranking criteria to enhance the generation quality (Wang et al., 2023b). In this study, we employ a reference-free metric, R$Q$UGE, in the re-ranking process to enhance question generation performance.

## 3 Dialogizer

Dialogizer is a novel framework designed to generate contextually relevant ConvQA datasets of high quality from textual sources. In addition to the simple dialog reconstruction (DR) task (§3.1), the framework incorporates two novel training methodologies: Question-Answer Matching (QAM) (§3.2) and Topic-aware Dialog Generation (TDG) (§3.3). During the inference phase, the textual passage is segmented into sentences that serve as answers, and the trained model autoregressively generates questions relevant to each answer to complete the ConvQA dataset (§3.4). Furthermore, to consistently generate stable and high-quality questions, Dialo-

gizer employs re-ranking through beam search during the inference phase, taking into account the contextual relevance of the generated questions (§3.5). Figure 2 provides an illustrative overview of our framework.

### 3.1 Dialog Reconstruction

Dialogizer is basically trained with the Dialog Reconstruction (DR) task proposed in Dai et al. (2022), which includes the random masking of one utterance $d_m(t) = (u_1, u_2, ..., u_{t-1}, \bullet, u_{t+1}, ..., u_T)$ in the dialog $d = (u_1, u_2, ..., u_t, ..., u_T)$, with the masked utterance denoted by the symbol $\bullet$. The objective is to reconstruct the missing utterance $u_t$. Specifically, we utilize the T5 (text-to-text transfer transformer) model (Raffel et al., 2020) to implement Dialogizer. Dialogizer can be described as a generative model characterized by parameters $\theta$, which define a probability distribution $p_\theta$ of the target utterance $u_t$ given the masked dialog $d_m(t)$. The following DR training objective is set:

$$\mathcal{L}_{DR}(\theta) = -\sum_{d \in \mathcal{D}} \mathbb{E}_{u_t \sim d}\Big[log p_\theta(u_t \mid d_m(t))\Big],$$

where $D$ is a dialog corpus.

The vanilla dialog inpainting model is trained exclusively in the DR task. However, our observations have confirmed that questions generated by this model exhibit low contextual relevance. Given that the answers are already well-defined as they

are extracted from the passage, generating questions that align well with the answer is a crucial concern. Consequently, we identify two primary factors that contribute to this phenomenon. Firstly, since the DR is simply designed to reconstruct masked utterances in the original dialogs, training the model exclusively on the DR task results in insufficient learning of question-answer alignment necessary for ConvQA dataset. Second, the original model's reliance on document title information as a prompt during the inference stage poses challenges in determining the specific information that must be conveyed within the generated question. In this work, we aim to overcome these challenges and enhance the contextual relevance of questions in ConvQA dataset generation.

## 3.2 Question-Answer Matching

To address the challenge of insufficient acquisition of question-answer alignment in vanilla dialog inpainting, we incorporate the Question Answer Matching (QAM) task into the Dialogizer training. This task aims to enhance the model's ability to interpret long-term dependencies in question-answer pairs.

Similar to the BERT (Devlin et al., 2018) pre-training technique known as next-sentence prediction, this task focuses on the binary classification of matching QA pairs. During the training phase, we utilize a negative sampling method to extract negative answers for specific questions in ConvQA dialogs. This method involves randomly sampling from the same dialog, which is challenging due to the similarity of context. Accordingly, the objective of QAM is to train the model to classify positive answers as *MATCH* and negative answers as *NOT MATCH*, as shown in Figure 2.

Therefore, the QAM training objective is to minimize the following loss function:

$$\mathcal{L}_{QAM}(\theta) = - \sum_{d \in \mathcal{D}} \mathbb{E}_{q_t \sim d} \Big[ logp_\theta(t_P \mid q_t, \ a_t^P)$$
$$+ logp_\theta(t_N \mid q_t, \ a_t^N) \Big],$$

where $D$ is a corpus of ConvQA dialogs, $q_t$ is a randomly sampled question from dialog $d$, $a_t^P$ is a positive answer corresponding to question $qt$, $a_t^N$ is a negative answer randomly sampled from the dialog $d$, and $t_P$ and $t_N$ represent positive and negative target texts, respectively. Given the T5 architecture's representation of input and output as text strings, the target output $t_P$ is set to *"The*

*answer matches the question"*, whereas $t_N$ is set to *"The answer does not match the question"*. These outputs serve as reference labels for the model to learn and classify alignment within given question-answer pairs.

## 3.3 Topic-aware Dialog Generation

In typical question-generation tasks, the primary objective is to determine *what to ask* and *how to ask* (Pan et al., 2019; Ghanem et al., 2022). However, when generating questions through this framework, the answers or contents of the questions are predetermined. Therefore, the model must be able to generate good questions by utilizing hints from the predetermined *what to ask*. To address this issue, Dai et al. (2022) incorporates the document title within the prompt during the inference phase. However, this approach yields unsatisfactory results as document titles are often excessively abstract. Consequently, the model fails to generate answer-relevant questions, instead producing overly general questions such as *"What is {document_title}?"* or *"Are there any other interesting aspects about this article?"*.

To reduce the reliance on document titles and enable the Dialogizer to generate contextually relevant questions, we facilitate knowledge acquisition through the Topic-aware Dialog Generation (TDG) task. We hypothesize that incorporating extracted keywords as hints about *what to ask* would enhance the generation of more specific and answer-relevant questions. To ensure that the extracted keywords are effectively incorporated into the prompt during the inference phase, we introduce a TDG task during the training phase to train the utilization of extracted keywords.

The TDG loss is computed as

$$\mathcal{L}_{TDG}(\theta) = - \sum_{d \in \mathcal{D}} \mathbb{E}_{q_t \sim d} \Big[ logp_\theta(q_t \mid d_m(t), \ k_t) \Big],$$

where $k_t$ denotes the keywords extracted from the answer. $k_t$ is obtained by applying a keyword extractor to the answer corresponding to $q_t$ and subsequently utilized in the prompt with the format *"$Keyword : k_t$"*.

Ultimately, we train our Dialogizer model by aggregating the losses using the following approach:

$$\mathcal{L} = \mathcal{L}_{DR} + \lambda_{QAM} * \mathcal{L}_{QAM} + \lambda_{TDG} * \mathcal{L}_{TDG}.$$

| Source Dataset | Framework | RQUGE† | USR-DR (c) | USR-DR (f) | QRelScore_LRM | QRelScore_GRG | GPT2 |
|---|---|---|---|---|---|---|---|
| Wikipedia | baseline | 2.7579 | 0.9270 | 0.6455 | 0.4655 | 0.5077 | 0.6046 |
| | Dialogizer[1] | **3.8303** | **0.9883** | **0.9416** | **0.5303** | **0.5893** | **0.6570** |
| Pubmed | baseline | 2.5406 | 0.9430 | 0.7170 | 0.4802 | 0.4589 | 0.3958 |
| | Dialogizer[2] | **3.4380** | **0.9908** | **0.9416** | **0.5343** | **0.4982** | **0.5200** |
| CC-news | baseline | 2.2645 | 0.8650 | 0.5380 | 0.4212 | 0.5211 | 0.4326 |
| | Dialogizer[3] | **3.5748** | **0.9644** | **0.8339** | **0.4887** | **0.6630** | **0.4696** |
| Elsevier | baseline | 2.4185 | 0.9592 | 0.5961 | 0.4430 | 0.4007 | 0.2542 |
| | Dialogizer[4] | **3.7803** | **0.9887** | **0.8670** | **0.5402** | **0.6081** | **0.4367** |

Table 1: Automatic evaluation results on datasets generated using the baseline dialog inpainting framework and our proposed Dialogizer framework, based on four source datasets. † : utilized for re-ranking. [1]WikiDialog2, [2]PubmedDialog, [3]CC-newsDialog, [4]ElsevierDialog

## 3.4 Inference : Autoregressive Generation

The inference process of the Dialogizer framework comprises transforming a passage into a ConvQA dialog. In the inference phase, our Dialogizer model fills in the masked utterances in the partial dialog $(s_p, \bullet, s_1, \bullet, s_2, ..., \bullet, s_T)$ for a given passage $(s_1, s_2, ..., s_T)$ with a prompt $s_p$ using the document title. We then add the keyword prompt "$Keyword : k(s_t)$" before the mask token, where $k(s_t)$ represents the keywords extracted from the answer utterance $s_t$. The red block in Figure 2 represents the keyword prompt. Additionally, the model generates utterances auto-regressively to avoid discrepancies with the DR approach that generates one utterance at a time.

Namely, Dialogizer generates $\hat{u}_1$ with $(s_p, "Keyword : k(s_1)", \bullet, s_1)$, and continues generating $\hat{u}_2$ with $(s_p, \hat{u}_1, "Keyword : k(s_2)", \bullet, s_2)$ to fill in all masks auto-regressively, thereby completing the partial dialog to $(s_p, \hat{u}_1, s_1, ..., \hat{u}_T, s_T)$. The overall inference process can be shown at the bottom of Figure 2.

## 3.5 Re-ranking with contextual relevance

In addition, we incorporate re-ranking (RR) in the inference phase to improve the contextual relevance of the generated questions. Relying solely on corpus statistics to generate the most likely output in the decoding stage does not guarantee contextual quality. To ensure the quality of the generated questions, we opt for a re-ranking process based on contextual relevance. Mohammadshahi et al. (2022) have shown that re-ranking with RQUGE increases contextual relevance and enhances correlation with human judgment in sentence evaluation. Therefore, we utilize RQUGE to re-rank candidate questions. Specifically, the model first generates a set of k-candidate questions using beam search. Then, the

model selects the most relevant question to both the passage and the answer based on RQUGE.

## 4 Experiments

We deploy the Dialogizer framework to generate four datasets, validating the quality of the datasets through diverse evaluations. The experimental details are provided in Appendix C.

## 4.1 Model implementation

We evaluate Dialogizer's performance by comparing it to the original dialog inpainting as a baseline, ensuring identical conditions. The baseline is implemented using the same backbone model and training dataset as Dialogizer for a fair comparison.

Both models utilize a T5-base (Raffel et al., 2020) as their backbone. We train both frameworks on two open-domain dialog datasets, namely Daily Dialog (Li et al., 2017) and Task Masker (Byrne et al., 2019), along with two ConvQA datasets, OR-QUAC (Qu et al., 2020) and QReCC (Anantha et al., 2020). For both models, we use four datasets for the dialog reconstruction task. For Dialogzier, we also use these two ConvQA datasets for QAM and TDG tasks during training. During the TDG training, keyword extraction is performed using the T5-based model developed by Pęzik et al. (2022).

## 4.2 Generated Datasets

Using Dialogizer, we generate four ConvQA datasets for use in experiments. These datasets are developed by leveraging four source-text datasets from diverse domains: Wikipedia, PubMed, CC-News (Hamborg et al., 2017), and Elsevier OA CC-By (Kershaw and Koeling, 2020). Each dataset is named after its corresponding source dataset, namely WikiDialog2, PubmedDialog, CC-newsDialog, and ElsevierDialog. Detailed statistics

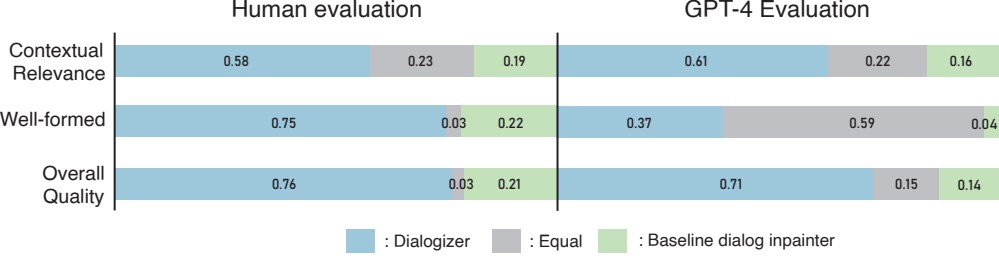

Figure 3: The human evaluation result comparing two datasets, each generated by the baseline dialog inpainting and our proposed Dialgozier. Ours obtained positive evaluations across all three criteria (contextual relevance, well-formedness, and overall quality), surpassing the baseline.

and sample dialogs for each dataset can be found in Appendices A and H.

### 4.3 Automatic Evaluation

**Evaluation Metrics** For quantitative comparison, we assess the generated datasets using diverse reference-free metrics that gauge distinct aspects of the dialog or generated questions. First, **RQUGE** is a reference-free metric for question generation that measures the quality of a given candidate question based on its corresponding answer and relevant passage. Similarly, **QRelScore** (Wang et al., 2022) is a context-aware evaluation metric for question generation that measures word- and sentence-level relevance without additional training or human supervision. This metric consists of QRelScore$_{LRM}$ and QRelScore$_{GRG}$: the former handles complex reasoning by calculating word-level similarity, while the latter measures factual consistency by comparing confidence in generating context. **USR-DR** (Mehri and Eskenazi, 2020) is a dialog evaluation metric specifically developed to evaluate the aspects of context maintenance, interest, and knowledge utilization through a retrieval task. This metric evaluates a dialog using retrieval results from the Ubuntu dialog corpus (Lowe et al., 2015), resulting in two categories: USR-DR($c$) incorporates history and facts, while USR-DR($f$) relies just on fact information for context. Pang et al. (2020) also proposed a **GPT-2 based dialog metric** that evaluates context coherence between sentences in a dialog.

**Results** We perform automatic evaluations by comparing the quality of the datasets produced by the baseline model and Dialogizer using four source datasets to demonstrate the effectiveness of the Dialogizer in generating ConvQA datasets. We present the main results in Table 1. Our findings demonstrate that the datasets generated by Dialogizer surpass those created by baseline across

all metrics. When evaluating the results for five metrics excluding R$Q$UGE, which is used for re-ranking, Dialogizer exhibits average performance improvements of 18.23% for Wikidialog2, 17.52% for PubmedDialog, 23.66% for CC-newsDialog and 38.80% for ElsevierDialog. The results validate that Dialogizer generates datasets of superior quality and enhanced contextual relevance compared to the baseline. The exceptional performance of Dialogizer across diverse source datasets from various domains further amplifies its value as a ConvQA dataset-generation framework.

### 4.4 Human and GPT-4 Evaluation

To comprehensively evaluate the quality of the questions generated by the baseline and Dialogizer, we randomly sample 100 dialogs from the generated datasets and conduct a relative comparison through both human and GPT-4. For both evaluations, we use three criteria, i.e., contextual relevance, well-formedness, and overall quality, according to the characteristics of the ConvQA task and existing research (Liang and Li, 2021). **Contextual relevance** measures the relevance of the question to the context and answer, **well-formedness** assesses whether the question is well-formed, and **overall quality** measures the overall quality of the context and question-answer pair. Human evaluation involves three crowd workers per question. Figure 3 shows that Dialogizer outperforms the baseline across all criteria. Additionally, when utilizing GPT-4 as an NLG evaluator (Wang et al., 2023a), our model consistently outperforms the baseline, confirming its superior performance. More detailed information regarding human and GPT-4 evaluations – inter-annotator agreement, payment details, instructions, and prompts – can be found in Appendices F and G.

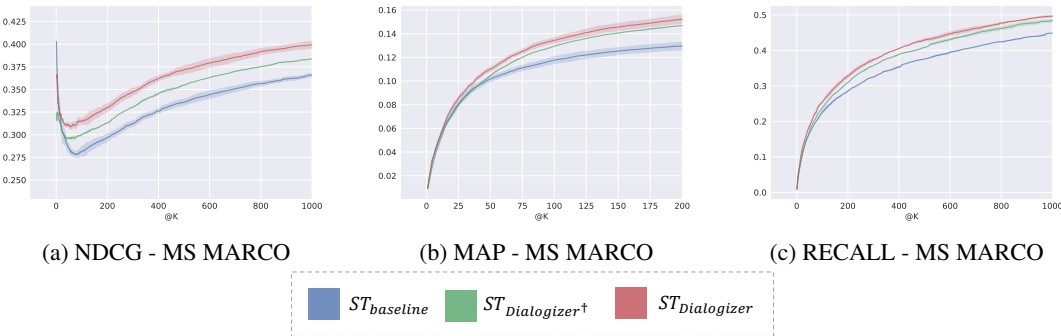

|  |  | (a) NDCG - MS MARCO |  |  | (b) MAP - MS MARCO |  | (c) RECALL - MS MARCO |

Figure 4: Average values for text retrieval benchmark of Sentence Trasnformer (ST) (Reimers and Gurevych, 2019) models trained on dialog datasets generated by three frameworks: baseline dialog inpainter, Dialogizer without re-ranking(†), and Dialogizer **(Higher is better)**. Shading indicates a standard deviation across five seeds.

| DR | QAM | TDG | RR | $RQUGE$ | USR-DR ($c$) | USR-DR ($f$) | QRelScore$_{LRM}$ | QRelScore$_{GRG}$ | GPT2 |
|----|-----|-----|----|---------|--------------|--------------|-------------------|-------------------|------|
| ✓ | - | - | - | 2.7579 | 0.9270 | 0.6455 | 0.4655 | 0.5077 | 0.6046 |
| ✓ | ✓ | - | - | 2.7818 | 0.9437 | 0.7308 | 0.4840 | 0.5643 | 0.6224 |
| ✓ | - | ✓ | - | 2.8732 | 0.9454 | 0.7381 | 0.4823 | 0.5283 | 0.6191 |
| ✓ | ✓ | ✓ | - | **2.9228** | **0.9472** | **0.7437** | **0.5003** | **0.5859** | **0.6232** |
| ✓ | - | - | ✓ | 3.6590 | 0.9585 | 0.7592 | 0.4895 | 0.5749 | 0.6257 |
| ✓ | ✓ | - | ✓ | 3.7095 | 0.9667 | 0.8141 | 0.5165 | **0.5923** | 0.6338 |
| ✓ | - | ✓ | ✓ | 3.7200 | 0.9753 | 0.8373 | 0.5210 | 0.5793 | 0.6394 |
| ✓ | ✓ | ✓ | ✓ | **3.8303** | **0.9883** | **0.9416** | **0.5303** | 0.5893 | **0.6570** |

Table 2: Results of an ablation study examining the impact of QAM, TDG, and Re-ranking (RR) components in the Dialogizer framework on the improvement of automatic evaluation performance. Wikipedia is used as a source dataset.

## 4.5 Application to Text Retrieval

In this section, we verify the alignment of the questions generated by our framework with the passage through a zero-shot text retrieval task. The coherence of query-passage pairs during training is crucial for improving performance in retrieving relevant information without explicit task training, especially in zero-shot scenarios. To gauge the alignment, we train a text retrieval model using the generated questions as queries paired with the original passage and assess its performance through a zero-shot text retrieval benchmark. We consider three experimental variations: baseline dialog inpainting, Dialogizer without re-ranking during the inference phase, and Dialogizer with re-ranking for detailed understanding. These three methods are applied to the Wiki corpus to construct passage-query pair datasets and generate 10k questions as queries for each method. These queries are then matched with their corresponding original passages to construct a training set for text retrieval.

Experimental results for the Ms-Marco benchmark (Nguyen et al., 2016) are shown in Figure 4.

The retrieval model trained on passage-question pairs generated by Dialogizer consistently outperforms those trained using the baseline in terms of NDCG (Järvelin and Kekäläinen, 2017), Mean Average Precision (MAP), and RECALL. Moreover, when Dialogizer incorporates re-ranking during the inference phase, it generates more relevant question pairs within the passage, further improving performance. These experimental results confirm that Dialogizer exhibits exceptional proficiency in generating questions relevant to a given passage. Experimental details and other benchmark results can be found in Appendix D.

## 5 Analysis

### 5.1 Ablation Study

To gain deeper insight into our method, we conduct an ablation study comparing the quality of datasets generated by applying each methodology to Wikipedia. As shown in Table 2, the Dialogizer model trained with TDG and QAM generates more contextually relevant questions, indicating that both proposed additional tasks effectively fa-

| Question Type | Proportion |
|---|---|
| **CONCEPT** 
 Asking for a definition or explanation of a concept | 38.49% |
| **EXAMPLE** 
 Asking for a examples of a concept | 32.91% |
| **VERIFICATION** 
 Seeking confirmation regarding truthfulness of a concept | 15.54% |
| **JUDGEMENTAL** 
 Requesting the answerer's own opinions | 8.57% |
| **COMPARISON** 
 Asking for comparison between multiple concepts | 4.49% |

Table 3: The distribution of question types in Wikidialog2, accompanied by explanations of their respective descriptions.

cilitate question-answer alignment learning. While TDG contributes slightly more to overall performance improvement, there are slight variations depending on the evaluation metric focused on assessing the different aspects of dialog quality. For instance, the R$Q$GUE and USR-DR metrics, which evaluate relevance by considering context, questions, and answers, indicate that TDG yields greater performance enhancement. In contrast, QRelScore, which evaluates question-answer pairs, demonstrates the effectiveness of QAM. The results obtained from training with both TDG and QAM simultaneously indicate that the two approaches synergistically contribute to generating more contextually relevant data. Furthermore, when performing re-ranking based on R$Q$GUE scores during inference, performance improvements are observed across all scenarios and metrics.

## 5.2 Question Types

We analyze the question-type distribution of open-ended questions in Wikidialog2. By referring to the 18 question types specified by Olney et al. (2012), we construct a question type ontology by merging ambiguous types (Cao and Wang, 2021). Table 3 shows that Wikidialog2 encompasses a diverse range of types, including 38% concepts, 31% examples, and 14% verifications. Regarding the ConvQA characteristic of information seeking, we note that there are relatively fewer judgemental questions that inquire about the respondent's opinion. Furthermore, as the original passage is segmented into sentence units to construct answers, it is inferred that fewer comparison-type questions that require comparing multiple concepts within a single question-answer pair. The experimental details can be found in the appendix E.

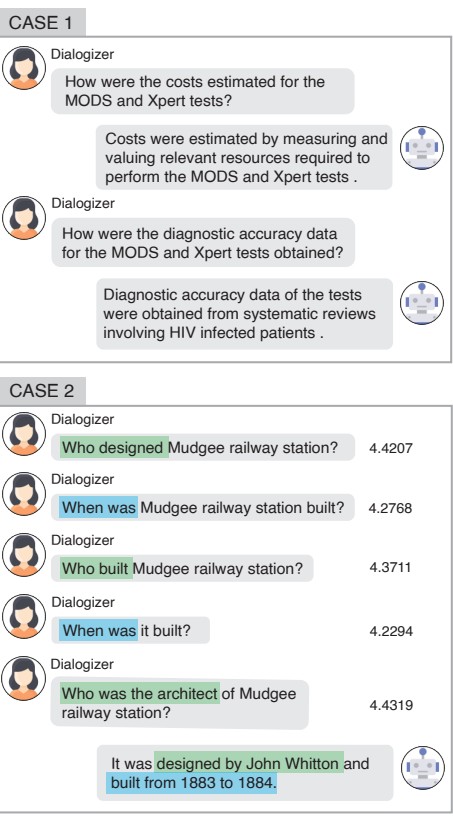

Figure 5: Case study. CASE 1 - Partial turns of the sample dialog from PubmedDialog. CASE 2 - Exploring diversity in generated questions with beam size 5.

## 5.3 Case Study

We present a sample application of Dialogizer in a specific domain and the diversity of multiple candidate questions generated using a beam search. Case 1 in Figure 5 presents a sample dialog extracted from PubmedDialog. Creating information-seeking dialog datasets in the medical domain is challenging because of the scarcity of experts. However, despite the absence of medical domain data in its training set, Dialogizer has demonstrated its ability to automatically generate high-quality medical dialog datasets. Case 2 in Figure 5 depicts sample candidate questions generated with a beam size of 5. These questions focus on various aspects of the answer and exhibit diverse expressions.

## 6 Conclusion

This study proposes Dialogizer, a framework that enables the automatic generation of high-quality dialog datasets from textual sources. Dialogizer is trained with the tasks of dialog reconstruction, question–answer matching for contextual alignment, and topic-aware dialog generation for spe-

cific question creation. Our experimental results demonstrate that Dialogizer produces contextually relevant ConvQA datasets. The proposed framework holds promise for advancing ConvQA research and its practical applications.

## Limitations

Our framework demonstrates improved performance compared to the baseline in terms of ConvQA dataset generation; however, it is computationally more expensive due to the inclusion of the beam search during the inference phase. When setting the beam size to 5, the inference time increased by approximately 2.4 times compared to greedy decoding. Since it is a dataset generation framework, the inference time may not be critical in real-world scenarios. However, when aiming to generate a large number of datasets for purposes such as data augmentation, the inference time should also be taken into consideration as a significant factor.

In addition, unlike the DR task, which can be applied to all dialogs datasets, the novel tasks proposed in this study, QAM and TDG, require the ConvQA dataset during the training process. This is because these tasks aim to effectively train the model in acquiring a comprehensive understanding of question-answer alignment, thus posing the limitation that well-matched question-answer pairs are necessary.

## Ethics Statement

To verify that the generated datasets do not contain any potential ethical problems, crowd workers were instructed to ascertain that the generated datasets do not include offensive, sexist, or racist comments; toxic language; or any instances of sexual behavior. The crowd workers were fairly compensated for their work. Additionally, a detailed description, interface to collect human evaluations, and further details of payment can be found in Appendix F.

Additionally, we utilize the GPT-4 model from official website of OpenAI* for GPT-4 evaluation. All models and datasets used in the experiments are from the publicly accessible website or Github repositories.

## Acknowledgements

This work was supported by LG AI Research. This work was partly supported by Institute of Information & communications Technology Planning & Evaluation (IITP) grant funded by the Korea government(MSIT) [NO.2021-0-01343, Artificial Intelligence Graduate School Program (Seoul National University) & NO.2021-0-02068, Artificial Intelligence Innovation Hub (Artificial Intelligence Institute, Seoul National University)], the BK21 FOUR program of the Education and Research Program for Future ICT Pioneers, Seoul National University in 2023, and the National Research Foundation of Korea (NRF) grant funded by the Korea government (No. 2021R1A2C2008855). K. Jung is with ASRI, Seoul National University, Korea.

---

*https://openai.com/

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

## A Generated Datasets Statistics

Table 4 shows the statistical information for the four datasets generated using Dialogizer, namely WikiDialog2, PubmedDialog, CC-newsDialog, and ElsevierDialog.

| Dataset | Source dataset | # of Dialogs | Average # of Turns |
|---------|----------------|--------------|--------------------|
| **WikiDialog2** | Wikipedia | 113,678 | 9.85 |
| **PubmedDialog** | Pubmed-writing | 22,811 | 9.24 |
| **CC-newsDialog** | CC-news | 69,846 | 10.49 |
| **ElsevierDialog** | Elsevier OA CC-By | 32,053 | 11.37 |

Table 4: Statistics of the four ConvQA datasets generated using the Dialogizer.

## B Reproductability checklists

### B.1 Dataset and Source code

We provide our experiment source code along with configuration code as supplementary materials. We will publicly release the generated datasets and the full codes with weight parameters.

### B.2 Computing Resources

Xeon 4210R (2.40 GHz) with RXT A6000 is used for the experiments. We use four GPUs for our experiments. All codes are implemented on Python 3.7.13 and PyTorch 1.10.1.

## C Dialogizer Training Details

### C.1 Training dataset

The statistics of the four training datasets used for baseline dialog inpainter and Dialogizer training can be found in Table 5.

| Dataset | # of Dialogs | Average # of Turns |
|---------|--------------|--------------------|
| **Daily Dialog** | 13,118 | 7.85 |
| **Task Master** | 54,255 | 19.34 |
| **OR-QuAC** | 5,644 | 14.36 |
| **QReCC** | 12,219 | 11.94 |

Table 5: Statistics of the four training datasets.

### C.2 Training configuration

We use T5-base model[†] as our Dialogizer model. The number of parameters of our model is about

---
[†]https://huggingface.co/t5-base

220M. The model trains with batch size 8 with gradient accumulation step size 8 and takes about 10 hours per epoch.

We use AdamW (Loshchilov and Hutter, 2017) optimizer with $\beta_1 = 0.9, \beta_2 = 0.999, \epsilon = 1e - 8$. The max gradient norm for gradient clipping is 1.0. In order to find the best-performing model, we conducted experiments on hyper-parameter combinations with 3 epoch steps: $\lambda_{QAM}$ : (0.05, 0.1, 0.5), $\lambda_{TDG}$ : (0.05, 0.1, 0.5), $per\_gpu\_batch\_size$ : (1, 2), $initial\_learning\_rate$ : ($1e - 4$, $5e - 5$, $2e - 5$), $warmup\_step$ : (0, 500). The hyper-parameter was manually tuned, and the best-performing model is with $\lambda_{QAM}$ 0.1, $\lambda_{TDG}$ 0.1, $per\_gpu\_batch\_size$ 2, $initial\_learning\_rate$ $5e - 5$, and $warmup\_step$ 0. We repeatedly conducted all experiments for four seed numbers.

## D Application to Text Retrieval Details

### D.1 Experiment Details

Figure 6 provides a detailed explanation of the experimental setup for text retrieval. The baseline dialog inpainter and Dialogizer model are employed to perform inference on the Wiki corpora, resulting in the creation of the WikiDialog dataset used for training the retrieval model. For Dialogizer, two versions of the WikiDialog dataset are generated based on the re-ranking criterion. Subsequently, the text retrieval model is trained using the three generated datasets: WikiDialog$_{baseline}$, WikiDialog$_{Dialogizer^†}$, and WikiDialog$_{Dialogizer}$.

We utilize the Sentence Transformer (ST) (Reimers and Gurevych, 2019) as the retrieval model and train it using the cosine similarity score as the ranking score through the MultipleNegativesRankingLoss. We also use AdamW optimizer and perform hyper-parameter tuning, as same in the Dialogizer training. The best-performing model is with $batch\_size$ 32, $initial\_learning\_rate$ $1e - 4$, $warmup\_step$ 0, and $num\_epochs$ 10. We repeatedly conduct all experiments for five seed numbers and report average values and standard deviation values.

### D.2 Metrics

We use three metrics for the text retrieval experiment: NDCG, MAP, and Recall. First, NDCG (Normalized Discounted Cumulative Gain) is a widely used metric that measures the quality of a ranked list of documents, considering both relevance and ranking position. MAP (Mean Aver-

age Precision) focuses on precision at different ranks and calculates the average precision across all queries, providing insights into the ability of a system to retrieve relevant documents. Finally, Recall measures the system's ability to retrieve all relevant documents, indicating the completeness of the retrieval process.

### D.3 Benchmarks

In addition to the Ms-Marco benchmark discussed in the main text, we conduct experiments on three additional benchmarks: Scifact (Wadden et al., 2020), Nfcorpus (Boteva et al., 2016), and Scidocs (Cohan et al., 2020).

### D.4 Results

The results for additional benchmarks in the text retrieval experiment can be found in Figure 7. Similar to MS-MARCO, across all benchmarks, $ST_{Dialogizer}$ demonstrates superior results in all metrics compared to $ST_{baseline}$, indicating that our methodology generates more coherent questions with the passage. Additionally, the effectiveness of re-ranking is further enhanced in these cases.

## E  Question Types Experiment Details

The question type analysis experiment is conducted on the RoBERTa-base (Liu et al., 2019) model, and the training dataset is created by Cao and Wang (2021). The model is trained using the Adam optimizer (Kingma and Ba, 2014), and the loss function used is CrossEntropyLoss. The best-performing classifier model is with $per\_gpu\_batch\_size$ 4, $learning\_rate$ $1e-5$, and $num\_epochs$ 5.

## F  Human Evaluation Details

The recruitment process for the three crowd workers for the purpose of human evaluation was conducted via the university's online community, specifically targeting individuals who possessed fluency in the English language. The crowd workers were provided with task definitions, instructions, and examples, as illustrated in Figure 8 and 9. Furthermore, they were notified that this evaluation is intended for academic purposes. After conducting the sample evaluation and calculating the required time, the crowd workers were fairly compensated to ensure a minimum hourly wage of $12 or higher, as calculated by the coworkers.

**Inter-Annotator Agreement (IAA)**  We measure the Inter-Annotator Agreement (IAA) among three crowd workers in the human evaluation process. We observe that Krippendorff's $\alpha$ and Cohen's kappa score indicate "substantial" or "moderate" agreement according to the referenced guideline (Landis and Koch, 1977). The Krippendorff's $\alpha$ and Cohen's kappa values for each of the three criteria are as follows:

**Contextual Relevance**
Kripependorff alpha: 0.617 (Substantial)
A1-A2 Cohen's kappa score: 0.661 (Substantial)
A1-A3 Cohen's kappa score: 0.613 (Substantial)
A2-A3 Cohen's kappa score: 0.537 (Moderate)
**Well-formed**
Kripependorff alpha: 0.654(Substantial)
A1-A2 Cohen's kappa score: 0.599 (Moderate)
A1-A3 Cohen's kappa score: 0.632 (Substantial)
A2-A3 Cohen's kappa score: 0.522 (Moderate)
**Overall Quality**
Kripependorff alpha: 0.630 (Substantial)
A1-A2 Cohen's kappa score: 0.599 (Moderate)
A1-A3 Cohen's kappa score: 0.646 (Substantial)
A2-A3 Cohen's kappa score: 0.547 (Moderate)
(A1,A2, and A3 stands for Annotator1, Annotator2, and Annotator3)

## G  GPT-4 Evaluation Details

The prompt used for GPT-4 evaluation is devised by referencing Liu et al. (2023), and the input template can be found in Table 6.

## H  Generated Dialog Examples

Sample dialogs for the four datasets created using Dialogizer (WikiDialog2, PubmedDialog, CC-newsDialog, and ElsevierDialog) can be found in Table 7-14.

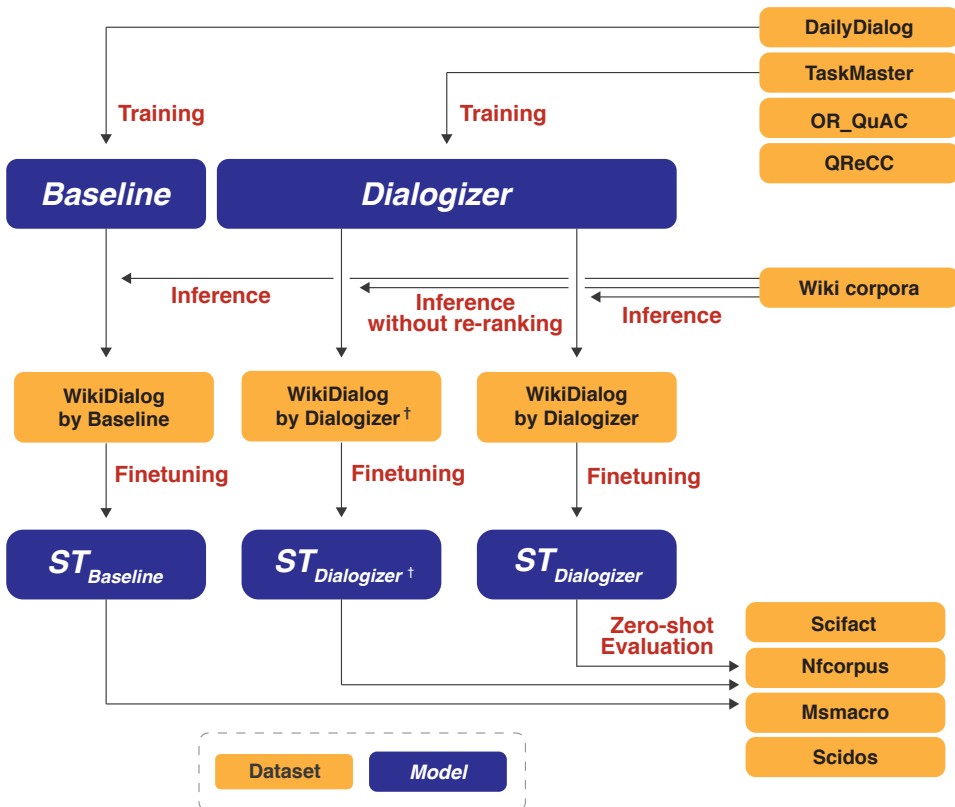

Figure 6: The overview of the text retrieval experiment.

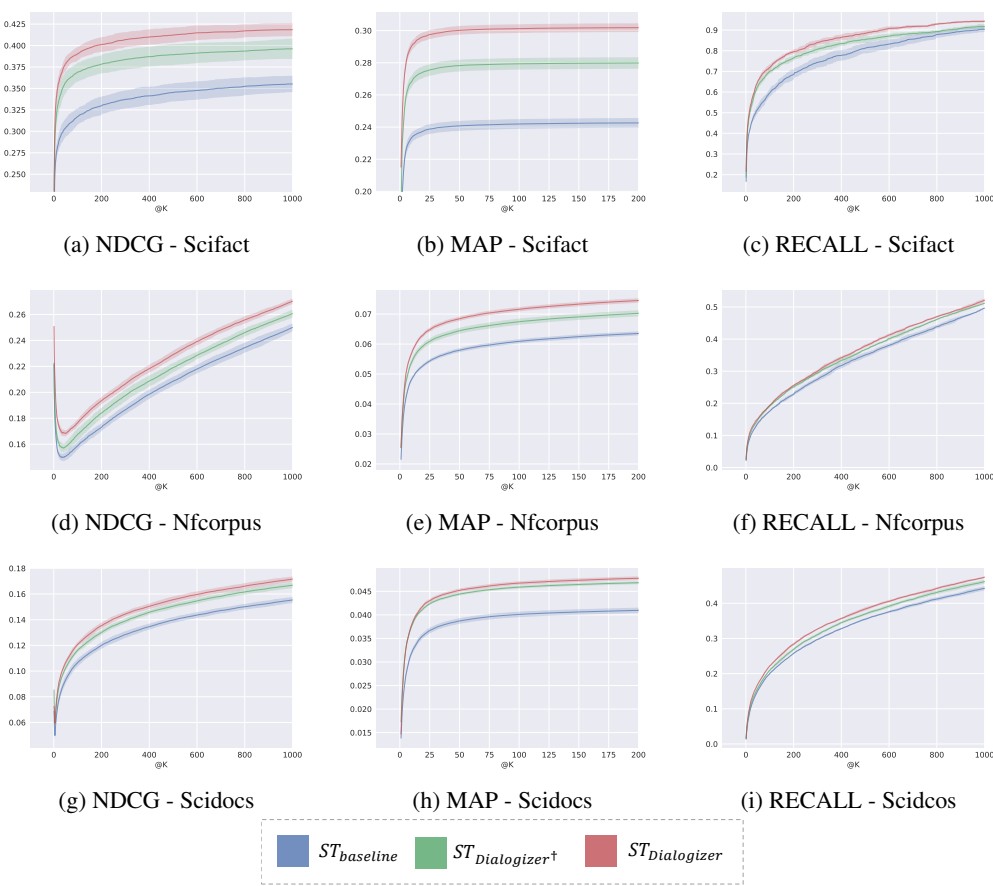

Figure 7: Average values for text retrieval benchmarks of Sentence Trasnformer (ST) (Reimers and Gurevych, 2019) models trained on dialog datasets generated by three frameworks: baseline dialog inpainter, Dialogizer without reranking(†), and Dialogizer **(Higher is better)**. Shading indicates a standard deviation across five seeds.

# Conversational Question Anwering Evaluation

## Instruction

This is a task to evaluate the quality of a **conversational question answering dataset**. You will be given [context, two candidate questions, answer], and your task is to compare the quality of the candidate questions based on four criteria: **contextual relevance, well formness, fluency, overall quality**. Please read the instructions carefully and make sure you understand them before proceeding with the task. If you have any questions, feel free to ask before continuing with the task. Please select equal **only** when it is difficult to judge.

- **Contextual Relevance**: whether the question relevant to the answer/context

- **Well-formedness**: whether the question is well-formed

- **Overall Quality**: overall quality of the question

---

**Context:** Another hot spring flowed out of the embankment at the Paso Robles Street exit on U.S. Route 101. Other unreinforced masonry buildings, some more than a century old, in the city's historic downtown area also had extensive damage. However, none of the buildings that had even partial retrofitting collapsed. There was a wrongful death lawsuit filed by the relatives of the 2 women killed in the earthquake against Mary Mastagni, and several trusts which owned the Acorn Building. The jury found Mastagni negligent in the care and maintenance of the Acorn Building, due to not retrofitting the building, in violation of city ordinances. The jury awarded nearly $2 million to the plaintiffs.

---

**Question A:** Is there a wrongful death lawsuit filed against Mary Mastagni and several trusts which owned the Acorn Building?

**Question B:** What happened after the 2003 San Simeon earthquake?

**Answer:** There was a wrongful death lawsuit filed by the relatives of the 2 women killed in the earthquake against Mary Mastagni, and several trusts which owned the Acorn Building.

Figure 8: Interface of human evaluation comparing the two methods: vanilla dialog inpainting and Dialogizer. (1/2)

Which question is more relevant to the given answer?

- ● Undecided
- ○ Question A
- ○ Equal
- ○ Question B

Which question is more well-formed?

- ● Undecided
- ○ Question A
- ○ Equal
- ○ Question B

Which question has better overall-quality?

- ● Undecided
- ○ Question A
- ○ Equal
- ○ Question B

Does the whole text (passage, questions, answer) has any potential risk? Potential risk means: (1. offensive, sexist, or racist comments, 2. toxic words, 3. sexual behavior.)

- ○ Yes
- ● No

If you answered 'Yes', please provide a brief explanation.

[                                                                    ]

Next Question

In order to proceed to the next question, you must answer all of the questions.

Figure 9: Interface of human evaluation comparing the two methods: vanilla dialog inpainting and Dialogizer. (2/2)

This is a task to evaluate the quality of a conversational question answering dataset. You will be given [context, two candidate questions, answer], and your task is to compare the quality of the candidate questions based on four criteria: contextual relevance, well-formedness, fluency, overall quality. For each criteria, answer which question is better.

1. **Contextual Relevance:** whether the question relevant to the answer/context
2. **Well-formedness:** whether the question is well-formed
3. **Overall Quality:** overall quality of the question

• Context:
• Question A:
• Question B:
• Answer:

Choose the question which is more relevant to the given answer.

options: [Question A, Equal, Question B]

Choose the question which is more well-formed?

options: [Question A, Equal, Question B]

Choose the question which has better overall-quality.

options: [Question A, Equal, Question B]

Table 6: The template of the prompt used for GPT-4 evaluation.

| | |
|---|---|
| USER: | **Where is Grevillea rudis found?** |
| AGENT: | Grevillea rudis Grevillea rudis is a shrub of the genus "Grevillea" native to an area along the west coast in the Wheatbelt region of Western Australia. |
| USER: | **How tall is the shrub?** |
| AGENT: | The loose, spreading to erect shrub typically grows to a height of and has non-glaucous branchlets. |
| USER: | **How do the leaves of the shrub Grevillea rudis look like?** |
| AGENT: | It has simple flat, spathulate, irregularly lobed leaves with a blade that is long and wide. |
| USER: | **How often does the shrub Grevillea rudis bloom?** |
| AGENT: | It blooms sporadically throughout the year and produces a terminal raceme regular inflorescence with cream or yellow flowers and white or cream styles. |
| USER: | **What kind of fruit does the shrub Grevillea rudis produce?** |
| AGENT: | Later it forms obovoid or ellipsoidal glandular hairy fruit that is long. |
| USER: | **How does the shrub Grevillea rudis regenerate?** |
| AGENT: | It will regenerate from seed only. |

Table 7: The first sample dialog extracted from WikiDialog2. The utterances generated by Dialogizer are expressed in bold.

| | |
|---|---|
| USER: | **Why does Freeview use the DVB-T ODFM standard?** |
| AGENT: | Freeview uses the DVB-T ODFM standard for terrestrial transmission, as established in 2001 with NZS6610:2001, to avoid the multipath problem caused by New Zealand's rugged topography. |
| USER: | **What is the rival US standard that uses 8-VSB modulation?** |
| AGENT: | ATSC, the rival US standard that uses 8-VSB modulation, had a number of first gen. |
| USER: | **What were the first gen demodulators?** |
| AGENT: | demodulators that couldn't handle multipath well, so it was not chosen. |
| USER: | **What is terrestrial Freeview broadcast in?** |
| AGENT: | Terrestrial Freeview is broadcast in H.264, which unlike H.262 has an expensive transmission patent licensing tax for free TV and subscription use. |
| USER: | **Did people who took part in the Auckland digital trial using terrestrial H.262 receivers have to change their receivers to more expensive H.264 models in order to receive terrestrial Freeview?** |
| AGENT: | People who took part in the Auckland digital trial using terrestrial H.262 receivers needed to change their receivers to more expensive H.264 models in order to receive terrestrial Freeview. |

Table 8: The second sample dialog extracted from WikiDialog2. The utterances generated by Dialogizer are expressed in bold.

| | |
|---|---|
| USER: | **How many hospitalizations does Orthostatic Hypotension cause?** |
| AGENT: | Orthostatic_hypotension causes ≈ 80000 hospitalizations per year in the United States . |
| USER: | **What are the treatment options for Orthostatic Hypotension?** |
| AGENT: | Treatments for Orthostatic_hypotension include Fludrocortisone , a mineralocorticoid analog that promotes sodium reabsorption ; and midodrine , an $\alpha$-1 adrenergic agonist that is a direct vasoconstrictor . |
| USER: | **How safe are both medications for Orthostatic Hypotension?** |
| AGENT: | Although both medications are used to treat Orthostatic_hypotension , few studies have compared their relative safety |
| USER: | **Are there any other interesting aspects about this article?** |
| AGENT: | We compared incidence rates of hospitalizations for all causes , and for congestive heart failure between users of Fludrocortisone and users of midodrine in a retrospective cohort study of Tennessee Medicaid adult enrollees ( 1995–2009 ) . |
| USER: | **How were the adjusted incidence rate ratios calculated?** |
| AGENT: | Adjusted incidence rate ratios were calculated using negative binomial regression models . |

Table 9: The first sample dialog extracted from PubmedDialog. The utterances generated by Dialogizer are expressed in bold.

| | |
|---|---|
| USER: | **What is osteoarthritis?** |
| AGENT: | Osteoarthritis is a degenerative joint disorder of articular cartilage and is the most common type of arthritis in the elderly . |
| USER: | **Is hydroxychloroquine used to treat osteoarthritis?** |
| AGENT: | There are only a few reports regarding the use of hydroxychloroquine in the treatment of Osteoarthritis . |
| USER: | **What is the effect of hydroxychloroquine on symptoms of knee-Osteoarthritis?** |
| AGENT: | To investigate the effects of hydroxychloroquine on the symptoms of mild to moderate knee_Osteoarthritis ( Kellgren and Lawrence grade II and III ) , we performed a double-blind , placebo-controlled study in 44 patients . |
| USER: | **How many hydroxychloroquine pills did the 44 patients receive?** |
| AGENT: | The patients were randomly assigned to two groups : one group received hydroxychloroquine pills ( 200 mg twice daily ) and the other group received placebo pills . |

Table 10: The second sample dialog extracted from PubmedDialog. The utterances generated by Dialogizer are expressed in bold.

| | |
|---|---|
| USER: | **How do you write bug-free software?** |
| AGENT: | Writing bug-free software is practically impossible, due to the impracticality of predicting every way in which code might be executed. |
| USER: | **What are some of the hidden flaws in the underlying programming language that can be exploited by hackers?** |
| AGENT: | But even if developers go above and beyond to avoid flaws that can be exploited by hackers, attackers can often still take advantage of vulnerabilities in the design of the underlying programming language. |
| USER: | **What happened at the recent Black Hat Europe conference?** |
| AGENT: | At the recent Black Hat Europe conference, IOActive security services revealed it had identified flaws in five major, interpreted programming languages that could be used by hackers in crafting an attack. |
| USER: | **What are the interpreted programming languages vulnerabilities?** |
| AGENT: | "With regards to the interpreted programming languages vulnerabilities, software developers may unknowingly include code in an application that can be used in a way that the designer did not foresee," it writes. |

Table 11: The first sample dialog extracted from CC-newsDialog. The utterances generated by Dialogizer are expressed in bold.

| | |
|---|---|
| USER: | **How many doctors are freed; strike continues?** |
| AGENT: | A Kenyan court has released seven doctors who are officials in the medics' union and who were jailed earlier this week for not calling off a strike by doctors working in public institutions. |
| USER: | **When did the court free the seven?** |
| AGENT: | The decision to free the seven was made Wednesday by three judges of the appellate court. |
| USER: | **How did the public react to the release of the seven?** |
| AGENT: | About 1,000 doctors outside the court celebrated the officials' release and held a peaceful march to Parliament and Nairobi's Freedom Park. |
| USER: | **Why are the doctors on strike?** |
| AGENT: | More than 5,000 doctors from public hospitals are on strike over pay and to protest Kenya's dilapidated health care system. |
| USER: | **What did the health minister say about the release?** |
| AGENT: | Health minister Dr. Cleopa Mailu told the Senate committee for health that he had agreed to the release of the officials. |
| USER: | **What did the union say about the release of the officials?** |
| AGENT: | The union had said that no negotiations would be held until the seven were freed. |

Table 12: The second sample dialog extracted from CC-newsDialog. The utterances generated by Dialogizer are expressed in bold.

| | |
|---|---|
| USER: | **What is a biomarker for anticoagulation?** |
| AGENT: | There is clinical need for a laboratory biomarker to identify patients who, following an unprovoked venous thrombosis (VTE), are at low VTE recurrence risk and can discontinue anticoagulation after a limited treatment duration (3–6 m). |
| USER: | **What is a secondary analysis of the ExACT study?** |
| AGENT: | This secondary analysis of the ExACT study aimed to evaluate whether quantitation of peripheral blood endothelial progenitor cells (EPCs) could improve prediction of VTE recurrence risk. |
| USER: | **Was the ExACT study a non-blinded, multicentre RCT?** |
| AGENT: | The ExACT study was a non-blinded, multicentre RCT comparing extended vs discontinued anticoagulation following a first unprovoked VTE. |
| USER: | **Who was eligible for the study?** |
| AGENT: | Adult patients were eligible if they had completed $\geq$ 3 months anticoagulation and remained anticoagulated. |
| USER: | **What was the primary outcome?** |
| AGENT: | The primary outcome was time to first recurrent VTE from randomisation. |
| USER: | **How long did the study follow up?** |
| AGENT: | Blood samples were taken at baseline and results correlated with clinical outcome over 2 years follow up. |

Table 13: The first sample dialog extracted from ElsevierDialog. The utterances generated by Dialogizer are expressed in bold.

| | |
|---|---|
| USER: | **What do B cells do?** |
| AGENT: | B cells constitute an essential line of defense from pathogenic infections through the generation of class-switched antibody-secreting cells (ASCs) in germinal centers. |
| USER: | **How do B cells start germinal center reactions?** |
| AGENT: | Although this process is known to be regulated by follicular helper T (TfH) cells, the mechanism by which B cells initially seed germinal center reactions remains elusive. |
| USER: | **What is the role of NKT cells in B cell immunity?** |
| AGENT: | We found that NKT cells, a population of innate-like T lymphocytes, are critical for the induction of B cell immunity upon viral infection. |
| USER: | **How do B cells priming by resident macrophages work?** |
| AGENT: | The positioning of NKT cells at the interfollicular areas of lymph nodes facilitates both their direct priming by resident macrophages and the localized delivery of innate signals to antigen-experienced B cells. |
| USER: | **How many IL-4-producing cells are in NKT cells?** |
| AGENT: | Indeed, NKT cells secrete an early wave of IL-4 and constitute up to 70% of the total IL-4-producing cells during the initial stages of infection. |
| USER: | **How is the requirement of this innate immunity arm conserved in Zika-virus-infected macaques?** |
| AGENT: | Importantly, the requirement of this innate immunity arm appears to be evolutionarily conserved because early NKT and IL-4 gene signatures also positively correlate with the levels of neutralizing antibodies in Zika-virus-infected macaques. |

Table 14: The second sample dialog extracted from ElsevierDialog. The utterances generated by Dialogizer are expressed in bold.