# OpenReview forum: "Dialogizer: Context-aware Conversational-QA Dataset Generation from Textual Sources"
_EMNLP/2023/Conference — EMNLP 2023 Main_

### Official Review · Reviewer_ja3s · 2023-08-04

**Typos Grammar Style And Presentation Improvements:** ROUGH score rather than RQUGH score i…
**Soundness:** 2

**Excitement:**

3: Ambivalent: It has merits (e.g., it reports state-of-the-art results, the idea is nice), but there are key weaknesses (e.g., it describes incremental work), and it can significantly benefit from another round of revision. However, I won't object to accepting it if my co-reviewers champion it.

**Paper Topic And Main Contributions:**

This study introduces Dialogizer as a novel framework for generating contextually relevant ConvQA datasets from textual datasets. The framework incorporates two training methodologies, in addition to dialog reconstruction, to address the limitation of generating contextually-irrelevant questions. These methodologies include a question-answer matching (QAM) task and a topic-aware dialog generation (TDG) task.

**Questions For The Authors:**

1. Do authors try instruction tuning to achieve the goal of the propoese objectives in the paper? In my opinion, insturction tuning may achieve comparable or better performances. The authors should compare their methods with instruction tuning.

2. I am confused about the section of Topic-aware Dialog Generation. How to extract topic words? How to construct the answer? Or how to contruct the question?

3. Do you conduct hyper-parameters analysis?

4. using GPT-4 to evaluate the generated texts are more reasonable.

**Reasons To Accept:**

The experimental results and analysis are extensive.

The proposed framework for generating CQA dataset seems reasonable.

**Reasons To Reject:**

The proposed methods have low impact on current LLM-based research community.  May chatgpt can construct such datasets by providing proper prompts and examplars?

I think ROUGH score can not full represent the quality of generated texts in introduction. The authors should present: 1. more evaluated metrics to validate the generated texts from Dialogizer and Dialogue Inpainter in different aspects. 2. quantitive evaluated analysis of texts from Dialogizer and Dialogue Inpainter rather than some specific examples. Selecting only a few examples lacks persuasiveness.

**Reproducibility:**

3: Could reproduce the results with some difficulty. The settings of parameters are underspecified or subjectively determined; the training/evaluation data are not widely available.

**Reviewer Confidence:**

3: Pretty sure, but there's a chance I missed something. Although I have a good feel for this area in general, I did not carefully check the paper's details, e.g., the math, experimental design, or novelty.

---

> ### Author Rebuttal · Authors · 2023-08-27
>
> First of all, thank you for the comprehensive input on our study.
> We will provide responses to each of the questions.
>
> ### **Response for Reason to Reject 1-1 - Have low impact on current LLM-based research community**
>
> There has been significant advancement and attention directed towards LLMs, which excel at learning and engaging in conversations across general domains. However, there remains a relative dearth in research and models focused on domain expert LLMs[1,2,3]. To create a Domain Expert LLM, the essential components are high-quality conversations and questions. By applying our methodology to papers or documents authored by experts, we can generate a substantial ConvQA dataset composed of high-quality questions and answers. Consequently, this approach could prove beneficial in the current LLM community for the creation of domain-specific expert LLMs. While the application of our methodology to LLMs might enable the generation of datasets with higher quality, it is important to note that such an endeavor could prove challenging at the academic level. Nevertheless, we believe that a notable advantage lies in the ability to swiftly create datasets for training LLMs on smaller scales.
>
> ### **Response for Reason to Reject 1-2 - Comparison with ChatGPT**
>
> As mentioned earlier, we believe that if we are able to create a high-quality ConvQA dataset using a model of approximately the same caliber as T5, there would be a clear advantage in terms of cost-effectiveness when compared to LLM. However, in alignment with your assertion, conducting additional experiments by supplying proper prompts or examples to ChatGPT, performing passage-to-dialog transformations, and subsequently comparing the quality with Dialogizer has been undertaken with the belief that this will offer supplementary insights.
>
> For the experiments, we employed ChatGPT Plus (GPT-4, August 3 version). Due to cost considerations, we were unable to generate as many samples as Dialogizer, and thus, after generating 100 samples, we evaluated with RQUGE and proceeded with a manual analysis. Having experimented with various prompts, we have attached the best-performing prompt below in the rebuttal.
>
> The sample dataset generated using GPT-4 achieved a RQUGE score of **2.6467**, which stands lower in comparison to the Dialogizer's performance. Upon conducting a direct analysis of the samples, it can be seen that  GPT-4 indeed excels in generating fine-grained and well-formed sentences; however, there is a noticeable diminishment in answer relevance that can be observed. Extracted from the dialog generated using a single passage, we present below examples of good QA pairs and bad QA pairs. The three versions of the complete dialog for this example, generated by Dialog Inpainter, Dialogizer, and ChatGPT, can be observed below in the rebuttal(above the prompt used for this experiment). Incorporating such content into the appendix could potentially offer readers supplementary insights. Also, in the future, exploring methods such as refining prompts or instruction tuning to leverage LLM for passage-to-dialog studies could potentially yield valuable insights.
>
> **Good Example (Generated by GPT-4)**
> ```
> Q: Can you tell me about the history of Czech immigration in Argentina?
> A: Czechs in Argentina The Czech immigration in Argentina began during the World War 1 and it was split in four periods.
> ```
> ```
> Q: That's really interesting. How did the motivations or characteristics of the immigrants change in the later periods?
> A: The first one is slightly before to the World War 1, the second one from 1920 to 1930, the third one during the World War 2 and the fourth one, the smallest in migration proportion, during 1990. During the first two periods, the immigration group was mainly integrated by workers and farmers motivated by economic reasons.
> ```
> **Bad Example (Generated by GPT-4)**
> ```
> Q: That's fascinating. How have the Czech immigrants contributed to Argentinian society?
> A: Argentina has the largest Czech community in Latin America.
> ```
> ```
> Q: Do they maintain strong ties with their Czech heritage, or have they mostly assimilated into Argentinian culture?
> A: Czechs settled mainly in Buenos Aires, Gran La Plata, Rosario and Chaco.
> ```
> ### **Response for Reason to Reject 2 - Needs for evaluation on diverse metrics and quantitative evaluated analysis**
>
> In the introduction section, we elucidated the motivation behind our research by highlighting the lower R**Q**UGE scores of datasets generated using the dialog inpainter. Subsequently, after introducing the Dialogizer, we indicated in lines 103-106 of the introduction that we conducted automatic evaluations using diverse dialog metrics. Furthermore, we underscored in the introduction our engagement in human and LLM evaluations, as well as an application to text retrieval tasks. While detailed experimental outcomes are not provided in the introduction, a concise summary of our final experimental results and their significance is encapsulated in lines 107-110 of the introduction.
>
> For a more detailed explanation of our experiments: we have evaluated our methodology across diverse aspects by leveraging several metrics, including QRelScore(LRM & GRG), USR-DR(c), USR-DR(f), and the GPT-2 based dialog metric, in addition to R**Q**UGE. This can be seen in the Experimental Section (**Section 4.3**), **Table 1** and **Table 2**. Additionally, we verify our excellence through Human evaluation and LLM-based evaluation (ChatGPT evaluation) results (**Section 4.4**) and application to text retrieval task (**Section 4.5**). Through conducting diverse quantitative and qualitative assessments that compare the Dialogizer with the Dialogue Inpainter, we have provided evidence that the proposed dialogizer framework is able to generate high-quality ConvQA datasets.
>
> ### **Response for Question 1 - Question about the instruction tuning**
>
> Engaging in passage-to-dialog using instruction tuning, as you suggested, could also serve as a valuable point of comparison. However, currently, the absence of a training set specifically tailored for LLM instruction tuning makes the implementation of such an experimental setting challenging[4,5]. Dialog Inpainter proposed a method for training passage-to-dialog without utilizing Supervised Fine-Tuning (SFT), and in this work, we further conducted research to enhance the quality by incorporating several methodologies. To achieve high-quality question generation through instruction tuning, the utilization of labeled datasets from relevant tasks is required. As of now, given the absence of such datasets, the methodology we employed seems to be the most suitable approach.
>
> ### **Response for Question 2 - Question about the Topic-aware Dialog Generation**
>
> TDG constitutes a task wherein keywords are extracted from answers, employed to formulate prompts, and models are trained to generate questions accordingly. By utilizing the extracted keyword from s1, denoted as k(s1), a prompt is constructed as $(s_{p}, "Keyword: k(s_1)", \bullet, s_1)$, following which $\hat{u}_1$ is generated. For keyword extraction, we employed the T5-based keyword extraction model proposed by Pezik et al. (2022)[6]. A detailed explanation of TDG can be found in **sections 3.3** Topic-aware Dialog Generation and **3.4** Inference: Autoregressive Generation. The model utilized for keyword extraction is provided in **section 4.1** Model Implementation.
>
> ### **Response for Question 3 - Question about hyper-parameters analysis**
>
> We conducted multiple experiments for hyper-parameter tuning, varying hyper-parameters such as $\lambda_{{QAM}}$ : (0.05, 0.1, 0.5) and $\lambda{_{TDG}}$ : (0.05, 0.1, 0.5). Elaborated details on hyper-parameter tuning and the best-performing hyper-parameters can be found in Appendix C.2 Training configuration.
>
> ### **Response for Question 4 - Question about using GPT-4 to evaluate the generated texts**
>
> In the LLM evaluation, we utilized ChatGPT Plus (GPT-4). We appreciate the clarification, and in order to ensure clarity in the final version, we will specify the version in a footnote.
>
> ### **Response for Typos Grammar Style and Presentation Improvements**
>
> R**Q**UGE[7] is a newly introduced reference-free metric that measures the answerability of questions with respect to the given context. Hence, in relation to the points, R**Q**UGE actually matches our intended purpose. A more explanation of R**Q**UGE can be found in Section 4.3 Automatic Evaluation - Evaluation Metrics.
>
> ### **In conclusion,**
>
> Thank you for your valuable feedback. We have provided responses to your concerns and questions. If your concerns are sufficiently addressed, we appreciate it if you would consider accepting our paper. Otherwise, please let us know about your remaining concerns.
>
>
> ___
> **Examples of ConvQA data samples generated by Dialog Inpainter, Dialogizer, and ChatGPT**
>
> While it is unfeasible to present all examples, we present the outcomes of our experimentation with the foremost passage of WikiDialog as a means to provide a maximally objective comparison.
>
> **#1. Dialog Inpainter (baseline)**
> ```
> Q: When did the Czechs settle in Argentina?
> A: Czechs in Argentina The Czech immigration in Argentina began during the World War 1 and it was split in four periods.
> Q: What led to the split?
> A: It is estimated that around 40,000 Czechs arrived to Argentina until 1970.
> Q: Are there any other interesting aspects about this article?
> A: Argentina has the largest Czech community in Latin America.
> Q: Where did the majority settle?
> A: Czechs settled mainly in Buenos Aires, Gran La Plata, Rosario and Chaco.
> Q: What period did they settle?
> A: There are four Czech immigration periods in Argentina.
> Q: What led to this?
> A: The first one is slightly before to the World War 1, the second one from 1920 to 1930, the third one during the World War 2 and the fourth one, the smallest in migration proportion, during 1990. During the first two periods, the immigration group was mainly integrated by workers and farmers motivated by economic reasons.
> ```
>
> **#2. Dialogizer (ours)**
> ```
> Q: What is the history of Czechs in Argentina?
> A: Czechs in Argentina The Czech immigration in Argentina began during the World War 1 and it was split in four periods.
> Q: How many Czechs arrived in Argentina?
> A: It is estimated that around 40,000 Czechs arrived to Argentina until 1970.
> Q: Is there a large Czech community in Argentina?
> A: Argentina has the largest Czech community in Latin America.
> Q: Where did the Czechs settle in Argentina?
> A: Czechs settled mainly in Buenos Aires, Gran La Plata, Rosario and Chaco.
> Q: How many immigration periods did the Czechs in Argentina have?
> A: There are four Czech immigration periods in Argentina.
> Q: What are the four Czech immigration periods?
> A: The first one is slightly before to the World War 1, the second one from 1920 to 1930, the third one during the World War 2 and the fourth one, the smallest in migration proportion, during 1990. During the first two periods, the immigration group was mainly integrated by workers and farmers motivated by economic reasons.
> ```
> **#3. ChatGPT (GPT-4)**
> ```
> Q: Can you tell me about the history of Czech immigration in Argentina?
> A: Czechs in Argentina The Czech immigration in Argentina began during the World War 1 and it was split in four periods.
> Q: What are the four periods you mentioned?
> A: It is estimated that around 40,000 Czechs arrived to Argentina until 1970.
> Q: That's fascinating. How have the Czech immigrants contributed to Argentinian society?
> A: Argentina has the largest Czech community in Latin America.
> Q: Do they maintain strong ties with their Czech heritage, or have they mostly assimilated into Argentinian culture?
> A: Czechs settled mainly in Buenos Aires, Gran La Plata, Rosario and Chaco.
> Q: Could you elaborate on what each of these four periods entails?
> A: There are four Czech immigration periods in Argentina.
> Q: That's really interesting. How did the motivations or characteristics of the immigrants change in the later periods?
> A: The first one is slightly before to the World War 1, the second one from 1920 to 1930, the third one during the World War 2 and the fourth one, the smallest in migration proportion, during 1990. During the first two periods, the immigration group was mainly integrated by workers and farmers motivated by economic reasons.
> ```
> **ChatGPT prompt and dialog generation process**
>
> **STEP 1**
> ```
> Please fill in only [BLANK] in the next dialogue.
>
> [START]
> A: [BLANK]
> B: Czechs in Argentina The Czech immigration in Argentina began during the World War 1 and it was split in four periods.
> [END]
> ```
> **STEP 2**
> ```
> Please fill in only [BLANK] in the next dialogue.
>
> [START]
> A: Can you tell me about the history of Czech immigration in Argentina?
> B: Czechs in Argentina The Czech immigration in Argentina began during the World War 1 and it was split in four periods.
> A: [BLANK]
> B: It is estimated that around 40,000 Czechs arrived to Argentina until 1970.
> [END]
> ```
> ...
>
> **[References]**
>
> [1] Peskoff, Denis, and Brandon M. Stewart. "Credible without Credit: Domain Experts Assess Generative Language Models." Proceedings of the 61st Annual Meeting of the Association for Computational Linguistics (Volume 2: Short Papers). 2023.
>
> [2] Ge, Yingqiang, et al. "Openagi: When llm meets domain experts." arXiv preprint arXiv:2304.04370 (2023).
>
> [3] Shen, Yongliang, et al. "Hugginggpt: Solving ai tasks with chatgpt and its friends in huggingface." arXiv preprint arXiv:2303.17580 (2023).
>
> [4] Ouyang, Long, et al. "Training language models to follow instructions with human feedback." Advances in Neural Information Processing Systems 35 (2022): 27730-27744.
>
> [5] Zhou, Wangchunshu, et al. "Controlled text generation with natural language instructions." arXiv preprint arXiv:2304.14293 (2023).
>
> [6] Pęzik, Piotr, et al. "Keyword extraction from short texts with a text-to-text transfer transformer." Asian Conference on Intelligent Information and Database Systems. Singapore: Springer Nature Singapore, 2022.
>
> [7] RQUGE: Reference-Free Metric for Evaluating Question Generation by Answering the Question (https://aclanthology.org/2023.findings-acl.428) (Mohammadshahi et al., Findings 2023)

---

### Official Review · Reviewer_D2VE · 2023-08-05

**Soundness:** 4

**Excitement:**

4: Strong: This paper deepens the understanding of some phenomenon or lowers the barriers to an existing research direction.

**Paper Topic And Main Contributions:**

This paper proposes Dialogizer, an inpainting model for the question generation given a passage. Previous methods suffer from low contextual relevance. To overcome this, the authors propose to incorporate two training tasks and re-ranking during the inference phase. Using this framework, four ConVQA datasets have been produced, which are of higher quality compared to the baseline dialog inpainting model. The evaluation has been done through automatic evaluation using diverse metrics and human evaluation.

**Reasons To Accept:**

1. This paper is well-motivated and there is clear discussion and demonstrations of the limitation of previous methods.

2. The proposed method is presented in a clear way.

3. There are comprehensive experiments to show the effectiveness of the proposed method.

**Reasons To Reject:**

I don't find any reason to reject this paper.

**Reproducibility:**

5: Could easily reproduce the results.

**Reviewer Confidence:**

1: Not my area, or paper was hard for me to understand. My evaluation is just an educated guess.

**Typos Grammar Style And Presentation Improvements:**

I suggest using \log in formulas for better visual effect.

L254. + should be -

---

> ### Author Rebuttal · Authors · 2023-08-27
>
> We would like to express our sincere gratitude for your thoughtful evaluation of our paper. We are thankful for the positive feedback and for considering our work favorably.
>
> As you pointed out, our research aims to establish a methodology for the automated construction of high-quality ConvQA datasets. To achieve this, we developed a framework called "Dialogizer" that integrates various methodologies and experimentally demonstrated the performance of the Dialogizer. We hold a positive outlook in envisioning that this study could be conducive to the ConvQA research field. Your constructive insights and suggestions for visual improvement would have greatly contributed to the refinement of our research. We will carefully consider your comments as we move forward with the final revisions.
>
> Thank you once again for your valuable input.

---

### Official Review · Reviewer_7ih2 · 2023-08-07

**Soundness:** 4

**Excitement:**

4: Strong: This paper deepens the understanding of some phenomenon or lowers the barriers to an existing research direction.

**Paper Topic And Main Contributions:**

The paper's topic is about addressing the data scarcity issue in Conversational Question Answering (ConvQA) by proposing a novel framework called "Dialogizer" that automatically generates ConvQA datasets with high contextual relevance from textual sources.

1. The paper introduces the concept of dialog inpainting as a method to generate ConvQA datasets from pre-existing text datasets. It segments the text dataset into sentence-level units, which are used as answers, and utilizes a trained dialog inpainter to generate questions corresponding to these answers, completing the conversation.

2. To overcome the limitation of contextually-irrelevant questions, the paper proposes a novel framework called "Dialogizer." The framework incorporates two training tasks: question-answer matching (QAM) and topic-aware dialog generation (TDG).

3. During the inference process of generating dialog from the passage, re-ranking is conducted using contextual relevance as a metric, ensuring the generation of high-quality questions.

**Reasons To Accept:**

1. The paper's results demonstrate that Dialogizer-generated datasets exhibit higher quality and context relevance compared to baseline methods. This finding indicates the potential impact of the proposed framework in advancing ConvQA research and potentially improving the performance of dialog systems in practical applications.

2. The paper acknowledges the limitation of generating contextually-irrelevant questions using traditional dialog inpainting methods. By proposing Dialogizer and incorporating specific training tasks to address this limitation, the paper demonstrates an effort to overcome challenges faced by existing approaches.

3. The paper supports its claims with both quantitative evidence, such as reference-free dialog metrics, and qualitative evidence through human evaluation. The combination of these methods enhances the credibility of the research outcomes.

**Reasons To Reject:**

1. While the paper proposes a novel framework called "Dialogizer" for generating contextually relevant ConvQA datasets, the contribution may be seen as incremental rather than transformative. The paper builds upon existing dialog inpainting methods and introduces additional training tasks to address the limitation of contextually-irrelevant questions.

2. The paper focuses solely on the task of generating ConvQA datasets and does not explore the broader impact of the proposed framework on dialog systems or its potential applications beyond dataset generation.

3. The paper claims that Dialogizer-generated datasets exhibit higher quality and context relevance. However, the paper lacks a detailed analysis of the dataset quality and potential biases introduced by the automatic generation process. Are there more examples that compare the proposed datasets with the prior existing ones?

**Reproducibility:**

4: Could mostly reproduce the results, but there may be some variation because of sample variance or minor variations in their interpretation of the protocol or method.

**Reviewer Confidence:**

4: Quite sure. I tried to check the important points carefully. It's unlikely, though conceivable, that I missed something that should affect my ratings.

---

> ### Author Rebuttal · Authors · 2023-08-28
>
> We sincerely appreciate your thorough examination of our paper. Your considerate evaluation, along with the constructive feedback and favorable appraisal of our efforts, is truly valued. We will provide responses to the inquiries you've requested and the questions you've posed.
>
> ### **Response for Reason to Reject 1 - More detailed analysis of the dataset quality**
>
> As you have pointed out, we have endeavored to assess the Dialogizer through both quantitative and qualitative evaluations. In addition, we conducted an analysis of the generated datasets through the **Section 5 Analysis**. In **5.1 Ablation Study(Table 2)**, we conducted an analysis of performance enhancements based on various methodologies we proposed. And in **5.2 Question Types(Table 3)**, we examined the distribution of question types in the generated dataset, WikiDialog2. Also, through **5.3 Case Study** and **Appendix H Generated Dialog Examples**, we strived to effectively showcase the quality of the datasets created using the Dialogizer.
>
> ### **Response for Reason to Reject 2 - More examples that compare two methods**
>
> We will provide a response to your inquiry regarding your request for further examples comparing dialogs generated by the Dialogizer and the Dialog Inpainter. The original paper's **Appendix H Generated Dialog Examples** contains instances of data created with the Dialogizer, and in addition, here we present the results of transforming the passages used in these examples into dialogs using the Dialog Inpainter. The first examples from each of the four source datasets are subjected to comparison. (Tables 7, 9, 11, and 13)
>
> **Configuration of the provided dialog examples**
>
> For a direct comparison with the Dialog Inpainter, in the case of the Wikipedia source dialog dataset, we examine the dialogues generated by WikiDialog as provided by [1], the dialogues generated through our implementation of the Dialog Inpainter, and the WikiDialog2 produced using the Dialogizer. (Please note that the initial Dialog Inpainter implementation from [1] employs T5-XXL, whereas our implementations for both the Dialog Inpainter and Dialogizer utilize T5-base as the underlying backbone model.) For the remaining source datasets (Pubmed, CC-news, and Elsevier), we provide a comparison between the datasets generated using our implemented Dialogizer and the Dialog Inpainter.
>
> **A brief interpretation of the provided dialog examples**
>
> From the examples below, it can be seen that the dialog data generated using the Dialogizer exhibits a higher degree of context relevance than those generated by Dialog Inpainter. Especially noteworthy is the case of WikiDialog; it is worth emphasizing that despite being generated by the Dialog Inpainter, which is implemented with T5-XXL as the backbone model, the Dialogizer utilizing T5-base as its backbone has demonstrated the ability to create ConvQA data of higher quality. Furthermore, as we have indicated in the paper, within the WikiDialog example (1-1), it can be observed that "What is {Document_title}?" (in this instance, the document title is "Grevillea Rudis") and the question like "Are there any other interesting aspects about this article?" are generated, illustrating the generation of overly general questions.
>
> As you have suggested, incorporating such additional comparative examples into the appendix could offer readers further insights and potentially enhance the credibility of the study. Your valuable feedback is greatly appreciated.
>
> ### **In Conclusion,**
>
> we would like to express our sincere gratitude for your comprehensive review of our paper. We will revise the paper according to your comments, including the inclusion of additional examples in the camera-ready version.
>
> ___
>
> **1-1. Framework: Dialog Inpainter implemented by [1] / Source Dataset: Wikipedia**
> ```
> Q: What is Grevillea rudis?
> A: Grevillea rudis Grevillea rudis is a shrub of the genus "Grevillea" native to an area along the west coast in the Wheatbelt region of Western Australia.
> Q: Are there any other interesting aspects about this article?
> A: The loose, spreading to erect shrub typically grows to a height of and has non-glaucous branchlets.
> Q: What about the leaves?
> A: It has simple flat, spathulate, irregularly lobed leaves with a blade that is long and wide.
> Q: Does it have any flowers?
> A: It blooms sporadically throughout the year and produces a terminal raceme regular inflorescence with cream or yellow flowers and white or cream styles.
> Q: What about fruit?
> A: Later it forms obovoid or ellipsoidal glandular hairy fruit that is long.
> Q: Can you tell me how to propagate?
> A: It will regenerate from seed only.
> ```
> **1-2. Framework: Dialog Inpainter implemented by this work / Source Dataset: Wikipedia**
> ```
> Q: What's Grevillea rudis?
> A: Grevillea rudis Grevillea rudis is a shrub of the genus "Grevillea" native to an area along the west coast in the Wheatbelt region of Western Australia.
> Q: How big is Grevillea rudis?
> A: The loose, spreading to erect shrub typically grows to a height of and has non-glaucous branchlets.
> Q: What does Grevillea rudis have?
> A: It has simple flat, spathulate, irregularly lobed leaves with a blade that is long and wide.
> Q: Are there any other interesting aspects about this article?
> A: It blooms sporadically throughout the year and produces a terminal raceme regular inflorescence with cream or yellow flowers and white or cream styles.
> Q: Does Grevillea rudis produce fruit?
> A: Later it forms obovoid or ellipsoidal glandular hairy fruit that is long.
> Q: Are there any other interesting aspects about Grevillea rudis?
> A: It will regenerate from seed only.
> ```
> **1-3. Framework: Dialogizer implemented by this work / Source Dataset: Wikipedia**
> Please see Table 7 in the paper.
>
> **2-1. Framework: Dialog Inpainter implemented by this work / Source Dataset: Pubmed**
> ```
> Q: What is Orthostatic_hypotension?
> A: Orthostatic_hypotension causes ≈ 80000 hospitalizations per year in the United States .
> Q: What is the treatment for Orthostatic_hypotension?
> A: Treatments for Orthostatic_hypotension include Fludrocortisone , a mineralocorticoid analog that promotes sodium reabsorption ; and midodrine , an α-1 adrenergic agonist that is a direct vasoconstrictor .
> Q: Are there any other interesting aspects about this article besides Fludrocortisone, a mineralocorticoid analog that promotes sodium reabsorption, and midodrine, an 1 adrenergic agonist that is a direct vasoconstrictor?
> A: Although both medications are used to treat Orthostatic_hypotension , few studies have compared their relative safety
> Q: Are there any other interesting aspects about this article?
> A: We compared incidence rates of hospitalizations for all causes , and for congestive heart failure between users of Fludrocortisone and users of midodrine in a retrospective cohort study of Tennessee Medicaid adult enrollees( 1995–2009 ) .
> Q: Are Fludrocortisone and midodrine safe?
> A: Adjusted incidence rate ratios were calculated using negative binomial regression models.
> ```
> **2-2. Framework: Dialogizer implemented by this work / Source Dataset: Pubmed**
> Please see Table 9 in the paper.
>
> **3-1. Framework: Dialog Inpainter implemented by this work / Source Dataset: CC-news**
> ```
> Q: How can I help you?
> A: Writing bug-free software is practically impossible, due to the impracticality of predicting every way in which code might be executed.
> Q: Are there any programming languages with hidden flaws vulnerable to hackers?
> A: But even if developers go above and beyond to avoid flaws that can be exploited by hackers, attackers can often still take advantage of vulnerabilities in the design of the underlying programming language.
> Q: Are there any other interesting aspects about the five programming languages with hidden flaws vulnerable to hackers?
> A: At the recent Black Hat Europe conference, IOActive security services revealed it had identified flaws in five major, interpreted programming languages that could be used by hackers in crafting an attack.
> Q: What are the five programming languages with hidden flaws vulnerable to hackers?
> A: "With regards to the interpreted programming languages vulnerabilities, software developers may unknowingly include code in an application that can be used in a way that the designer did not foresee," it writes.
> ```
> **3-2. Framework: Dialogizer implemented by this work / Source Dataset: CC-news**
> Please see Table 11 in the paper.
>
> **4-1. Framework: Dialog Inpainter implemented by this work / Source Dataset: Elsevier**
> ```
> Q: What can you tell me about Anticoagulation?
> A: There is clinical need for a laboratory biomarker to identify patients who, following an unprovoked venous thrombosis (VTE), are at low VTE recurrence risk and can discontinue anticoagulation after a limited treatment duration (3–6 m).
> Q: What is the secondary analysis of the ExACT study?
> A: This secondary analysis of the ExACT study aimed to evaluate whether quantitation of peripheral blood endothelial progenitor cells (EPCs) could improve prediction of VTE recurrence risk.
> Q: What was the primary analysis of the ExACT study?
> A: The ExACT study was a non-blinded, multicentre RCT comparing extended vs discontinued anticoagulation following a first unprovoked VTE.
> Q: Are there any other interesting aspects about this article?
> A: Adult patients were eligible if they had completed ≥ 3 months anticoagulation and remained anticoagulated.
> Q: How was the primary outcome of the ExACT study?
> A: The primary outcome was time to first recurrent VTE from randomisation.
> Q: What else is about ExACT study?
> A: Blood samples were taken at baseline and results correlated with clinical outcome over 2 years follow up.
> ```
> **4-2. Framework: Dialogizer  implemented by this work / Source Dataset: Elsevier**
> Please see Table 13 in the paper.
>
> [References]
>
> [1] Dai, Zhuyun, et al. "Dialog inpainting: Turning documents into dialogs." International Conference on Machine Learning. PMLR, 2022.

---

### Official Review · Reviewer_d7bR · 2023-08-08

**Typos Grammar Style And Presentation Improvements:** Figure 3 caption dialgozier, Figure 4…
**Soundness:** 4

**Excitement:**

3: Ambivalent: It has merits (e.g., it reports state-of-the-art results, the idea is nice), but there are key weaknesses (e.g., it describes incremental work), and it can significantly benefit from another round of revision. However, I won't object to accepting it if my co-reviewers champion it.

**Missing References:**

2.2 only discusses a single related work (Dai et al 2022). There's a line of previous work on conversational question generation or dialog generation for data augmentation, which should be discussed in this section, not limited to the papers below:

Kim et al, 2022, Generating Information-Seeking Conversations from Unlabeled Documents
Gao et al, 2019, Interconnected Question Generation with Coreference Alignment and Conversation Flow Modeling
Gu et al, 2021, ChainCQG: Flow-Aware Conversational Question Generation

**Paper Topic And Main Contributions:**

This paper aims to improve dialog inpainting, a conversational QA pair generation method, with the goal of generating questions that are more relevant to the conditioning answers. The proposed method generally follows dialog inpainting, training with dialog reconstruction, while introducing two mechanisms to guide generation: question-answer matching, to discern relevant QA pairs; and topic-aware dialog generation, conditioning on extracted keywords. Automatic metrics, human and LLM evaluations were performed, and application to text retrieval tasks were also shown. The main contributions of this paper are:
1. Proposed an improved dialog inpainting method, using two additional mechanisms to guide conversational QA pair generation in order to obtain higher quality and more relevant questions.
2. Showed the inpainted datasets can be utilized for application to text retrieval tasks.

**Questions For The Authors:**

1. Since the focus of the proposed method is on ConvQA,  why evaluate on general non-conversational retrieval, unlike previous dialog inpainting work (Dai et al 2022)?
2. For the motivation of the paper, in the intro and in 3.1 it is stated that "from observation" the original inpainting method produces less relevant questions. In the intro it is then stated that the baseline has low RQUGE. However this is compared to the proposed method. Is it possible to quantify this without comparing to the proposed method (as this is the motivation)? This way the motivation would be more convincing.

**Reasons To Accept:**

Proposed method improves over vanilla inpainting, and experiments are quite thorough.

**Reasons To Reject:**

It seems from ablation study (table 2) that a lot of the gain (for automatic evaluation) can be attributed to re-ranking the final QA pair candidates with RQUGE, not from the contribution made by this paper (QAM and TDG). Human/LLM evaluation for the proposed method without re-ranking was not reported.
Missing discussion and comparison against previous work on conversational question generation.

**Reproducibility:**

3: Could reproduce the results with some difficulty. The settings of parameters are underspecified or subjectively determined; the training/evaluation data are not widely available.

**Reviewer Confidence:**

3: Pretty sure, but there's a chance I missed something. Although I have a good feel for this area in general, I did not carefully check the paper's details, e.g., the math, experimental design, or novelty.

---

> ### Author Rebuttal · Authors · 2023-08-27
>
> First of all, thank you for the thorough feedback on our study.
> We will offer explanations, individually addressing the queries and questions you hold.
>
> ### **Response for Reason to Reject 1 - Performance enhancement by re-ranking process**
> Our research aims to establish a robust framework for generating high-quality ConvQA datasets, wherein the re-ranking process during the inference stage stands as a component of our devised framework, alongside QAM and TDG, aimed at enhancing the quality of the generated questions. Furthermore, upon examining the outcomes of metrics (besides RQUGE [1]; directly in the re-ranking process), the performance of the results obtained by the simple re-ranking and the QAM + TDG demonstrates a similarity in their effectiveness compared to the baseline(see 2.  vs 3. in the table below), showcasing comparable levels of achievement. In addition, it is noteworthy that the combined application of re-ranking and QAM + TDG synergistically yielded additional performance enhancements(see 4. in the table below).
>
> |            | USR-DR (c) | USR-DR (f) | QRelScore(LRM) | QRelScore(GRG) | GPT2    |
> |------------|------------|------------|--------------|--------------|---------|
> | 1. baseline   | 0.9270     | 0.6455     | 0.4655       | 0.5077       | 0.6046  |
> | 2. QAM+TDG | 0.9472     | 0.7437     | **0.5003**       | **0.5859**       | 0.6232  |
> | 3. simple RR | **0.9585**     | **0.7592**     | 0.4895       | 0.5749       | **0.6257**  |
> | 4. QAM+TDG+RR | 0.9883     | 0.9416     | 0.5303       | 0.5893       | 0.6570  |
>
> (This table has been excerpted from Table 2 in the paper.)
>
> ### **Response for Reason to Reject 2 - Needs for human/LLM evaluation for the proposed method without re-ranking**
> As previously mentioned, re-ranking, being a component of the framework proposed in this paper, we conducted human and LLM evaluations with a dataset that integrates the entirety of the proposed framework. However, in alignment with your assertion, conducting a supplementary experiment by contrasting the dataset generated solely through the application of TDG+QAM with the Dialog Inpainter[2] could provide further insights through additional analysis. Therefore, we proceeded to conduct the following supplementary experiments.
>
> Due to time constraints, human evaluation could not be included in this rebuttal submission. The outcomes of the LLM evaluation comparing the TDG+QAM and the baseline can be observed below. The experimental settings remained consistent with those detailed in the paper.
> ```
> Dialogizer (with re-ranking) vs. baseline - Contextual Relevance
> 0.61 (D) 0.22(E) 0.16(B)
> Dialogizer (without re-ranking) vs. baseline - Contextual Relevance
> 0.52(D) 0.24(E) 0.22(B)
>
> Dialogizer (with re-ranking) vs. baseline - Well-formed
> 0.37(D) 0.59(E) 0.04(B)
> Dialogizer (without re-ranking) vs. baseline - Well-formed
> 0.35(D) 0.60(E) 0.05(B)
>
> Dialogizer (with re-ranking) vs. baseline - Overall Quality
> 0.71(D) 0.15(E) 0.14(B)
> Dialogizer (without re-ranking) vs. baseline - Overall Quality
> 0.60(D) 0.19(E) 0.21(B)
> ```
> (where D stands for Dialogizer, E stands for Equal, and B stands for Baseline[2])
>
> The model trained with QAM+TDG, while not surpassing that of Dialogizer with re-ranking, still showcases a significant enhancement compared to the baseline. Comparing the two versions of Dialogizer (with and without re-ranking), it can be observed that the version without re-ranking exhibits a slightly reduced degree of performance improvement in terms of Contextual Relevance and Overall Quality while showing nearly similar results in Well-formedness. This can be attributed to the metric used for re-ranking, RQUGE, which considers contextual relevance.
>
> It is deemed that the inclusion of the aforementioned experimental results and analysis in the final version would provide readers with additional insights.
>
> ### **Response for Question 1 - Question about application to text retrieval task (Section 4.5)**
>
> In contrast to the approach taken by [2](dialog inpainter), where the focus lay on proposing the passage to dialogue framework and demonstrating its utility by conducting experiments to establish the dataset's effectiveness in the context of ConvQA, our endeavor was driven by the objective of enhancing the quality of the generated ConvQA dataset. As such, we structured our experiments to validate this pursuit. Particularly, in order to assess the degree of answer relevance exhibited by the generated questions, we conducted experiments by applying them to a retrieval task[3]. The experimental setup of [2] involves generating an extensive amount of ConvQA data for pretraining to showcase performance enhancements, which certainly demonstrates the utility of the dataset. However, it falls short of focusing on confirming the quality of the generated questions.
>
> ### **Response for Question 2 - Question about the motivation of this work**
>
> The motivation behind this research was driven by the belief that the study conducted by [2] held great promise for advancing the field of ConvQA. However, upon personally examining the WikiDialog dataset released in their paper, the quality of the generated questions was not as high as initially anticipated. Upon manually examining a corpus of 100 sample question-answer pairs, we observed that more than 30 questions demonstrated a deficiency in maintaining a satisfactory level of context relevance. For instance, as demonstrated by the examples below, a substantial number of questions lacking contextual relevance were observed (drawn from various instances of WikiDialog). Also, overly general questions such as 'What is {document_title}?' or 'Are there any other interesting aspects about this article?' recur persistently.
>
> ```
> Q: Do any other characters die?
> A: Mary Alice Young (Brenda Strong) is seen in this episode.
> ```
> ```
> Q: Who did she manage to beat?
> A: Jennifer Maguire is a 27-year-old marketing consultant, originally from Dublin, who said that when she first ran an office, she made £60,000 in six months with no training.
> ```
> ```
> Q: What are some of the songs on the album?
> A: While the album was praised for its top production quality, complaints were often made about the overuse of either vocoders or the auto-tune software.
> ```
> ```
> Q: what is the difference between common ground and joint ventures?
> A: " Finding common ground" is a technique people use to facilitate interpersonal relationships.
> ```
> Therefore, we proceeded to quantitatively assess the dataset using the RQUGE, a reference-free metric known for its strong correlation with human judgment. WikiDialog achieved an approximate score of 2.7. Given that the range of RQUGE spans from 0 to 5 points, this result suggests considerable room for enhancement from the 2.7 score. Furthermore, building upon the evidence of identified subpar samples, we embarked upon a research endeavor focused on developing a higher-quality ConvQA dataset generation framework.
>
> For the purpose of reference, we provide examples of questions generated by **Dialogizer** utilizing the same passages of the aforementioned examples.
> ```
> Q: Who is seen in this episode?
> A: Mary Alice Young (Brenda Strong) is seen in this episode.
> ```
> ```
> Q: How did Jennifer Maguire start her career?
> A: Jennifer Maguire is a 27-year-old marketing consultant, originally from Dublin, who said that when she first ran an office, she made £60,000 in six months with no training.
> ```
> ```
> Q: Was there any criticism of the album?
> A: While the album was praised for its top production quality, complaints were often made about the overuse of either vocoders or the auto-tune software.
> ```
> ```
> Q: What is finding common grounds?
> A: " Finding common ground" is a technique people use to facilitate interpersonal relationships.
> ```
>
> ### **In conclusion,**
>
> once again, we greatly appreciate your detailed comments. If your issues have been adequately resolved, we would be grateful if you could contemplate accepting our paper. Otherwise, please let us know about any remaining concerns you may have. We will revise the paper in the final draft according to your comments, including suggestions for improvement, missing references, and typos.
>
> [References]
>
> [1] RQUGE: Reference-Free Metric for Evaluating Question Generation by Answering the Question (https://aclanthology.org/2023.findings-acl.428) (Mohammadshahi et al., Findings 2023)
>
> [2] Dai, Zhuyun, et al. "Dialog inpainting: Turning documents into dialogs." International Conference on Machine Learning. PMLR, 2022.
>
> [3] Mao, Yuning, et al. "Generation-augmented retrieval for open-domain question answering." arXiv preprint arXiv:2009.08553 (2020).

---

### Meta-Review · Area_Chair_esrA · 2023-10-04

**Recommendation:** 4

**Metareview:**

A dialog inpainting pipeline to convert dialogs into question-answering. Sound fidelity evaluation.

---

### Decision · Program_Chairs · 2023-10-07

**Decision:**

Accept-Main

**Comment:**

A dialog inpainting pipeline to convert dialogs into question-answering. Sound fidelity evaluation.